# Factors Related to the Compliance of Arab Parents in Israel to the Vaccination of Children and Adolescents against COVID-19

**DOI:** 10.3390/vaccines11101540

**Published:** 2023-09-28

**Authors:** Ola Ali-Saleh, Mohammad Khatib, Salam Hadid, Kamal Dahamsheh, Fuad Basis

**Affiliations:** 1The Max Stern Yezreel Valley College, Emek Yezrael 1930600, Israel; olaa@yvc.ac.il; 2The Galilee Society, Zefat Academic College, University of Haifa, Zefat 13206, Israel; khatib.health@gmail.com; 3Zefat Academic College, Safed 1320611, Israel; 4Nazareth Academic School of Nursing, Nazareth 16100, Israel; salam_hadid@nazhosp.com (S.H.); kamaldahamsheh@nazhosp.com (K.D.); 5Rambam Health Care Campus, Haifa 3109601, Israel

**Keywords:** Arab, Israel, minority, children, adolescents, COVID-19 vaccine

## Abstract

The Arab minority consists of 20% of the Israeli population. Most of the Arab minority live in rural villages, mostly in closed communities, with specific psycho-social and sociodemographic characteristics. Previous studies showed different attitudes to COVID-19 vaccination among Arab adults. Objectives: To examine the influence of factors on the willingness of parents to vaccinate their young and adolescent children and if there is a difference between parents’ attitudes to both groups. Methods and Materials: The survey examined correlations between variables and attitudes toward the vaccine and the pandemic and was disseminated anonymously to parents of children between the ages of 5 and 18 in the Arab population of Israel through social media networks, using the snowball method. Statistical analyses included the Pearson correlation, MANCOVA, and logistic regression tests. Results: In total, 361 Arab Israeli parents participated. As mentioned above, 130 parents had both children and adolescents. Overall, 48 parents (36.9%) chose not to vaccinate both their children and adolescents, 52 parents (40.0%) chose to vaccinate only their adolescents, only 1 parent (0.8%) chose to vaccinate only the child, and 29 parents (22.3%) vaccinated both their children and adolescents. Significant correlations were found among a higher age of parents and socioeconomic status, attitudes toward COVID vaccination, subjective norms, perceived severity of the disease, perceived benefits of vaccination, and trust in formal sources. Discussion: There is a difference between parents’ decision to be vaccinated and their willingness to vaccinate their children. There is a difference between their decision to vaccinate their adolescents and their young children. Different factors positively or negatively influenced parents’ decisions. Addressing these factors by authorities may increase compliance of Arab minorities with instructions in the future.

## 1. Introduction

On 11 March 2020, the World Health Organization (WHO) declared COVID-19 a pandemic [1]. Due to the high transmission rates and increased morbidity and mortality, efforts were made by researchers to develop vaccines to stop the spread of the virus [2].

In December 2020, Israel started administering BNT162b2, an anti-COVID-19 mRNA-based vaccine developed by BioNTech and Pfizer. Priority was given to seniors, that is citizens aged 60 years and above; gradually, the age limit was reduced to allow younger adults (those aged at least 18 years) to be vaccinated [3].

In April 2021, the Israeli Ministry of Health (MOH) approved vaccination against COVID-19 for children aged at least 12. The vaccination campaign continued through the summer to enable 70% of pupils and students to return to schools and universities [4].

On 29 October 2021, the FDA approved the COVID-19 vaccine for use by children aged 5–11 years old [5,6]. COVID-19 vaccination for children was approved by Israel’s MOH on 14 November 2022 [7]. The national health maintenance organizations (HMOs) and medical services opened COVID-19 vaccination clinics for children on 23 November 2022. There were campaigns to encourage parents and children to get vaccinated. The strategies used by the MOH involved the recruitment of professionals, including pediatricians, to present information on TV and the MOH website (“Pediatric COVID-19 vaccination”) [8]. The recruitment of medical doctors to educate the public through the media proved effective because of the high trust in pediatricians who were perceived by the parents as being truly caring toward children. In addition, an open letter to parents, signed both by the Ministry of Education’s and MOH’s general managers, assured the parents of the safety of the vaccine and encouraged them to vaccinate their offspring [9]. Moreover, public health nurses administered the vaccine within school premises to children whose parents had signed a consent form.

A survey by the Shiluv Institute for Family and Couple Therapy, Jerusalem, showed that the main reason that motivated adult Arabs to get vaccinated was primarily to acquire a “vaccination certificate,” since the certificate served as an entrance ticket to certain places, such as working places, seniors’ homes, and airports, as well as a ticket to travel abroad. A second factor that encouraged people to get vaccinated was the assurance by the MOH that the vaccine is not dangerous in the long run [7].

The survey findings also showed that most Arab parents were willing to vaccinate their children aged 12–17 and 5–11 years old (about 67% and 62%, respectively) and indeed booked an appointment for the vaccination. Most people in Israel were convinced that pediatricians do care about children’s wellbeing and that the MOH is concerned about public health (about 78% and 73%, respectively). However, 43% of the Arabs surveyed believed that Israel was being used as an “experimental laboratory” by Pfizer to test the efficacy and safety of the vaccine; 25% believed that children vaccination was performed without caution for economic reasons, i.e., the authorities wanted to vaccinate the children as soon as possible so that they could return to school and their parents could return to work and promote the financial situation of the country. Finally, 75% of the public reported being pressured by community members to vaccinate their children [7].

Data on child COVID-19 vaccination reported in January 2022 revealed that among children aged 12–15 years, Denmark led with 62.7% of teenagers who had received two doses of the COVID-19 vaccine, while Israel was second with 57.4%. In the same report, Israel was in the sixth position for vaccination of children aged 5–11 years at 21.7%, after Spain, Iceland, Portugal, Denmark, and Austria (47.1%, 45.8%, 41.1%, 40.8%, and 22.6%, respectively) [7].

According to an Israeli MOH report, the rate of adverse events following COVID-19 vaccination among children aged 5–11 is lower than that found in clinical trials. The adverse events reported mostly included general symptoms such as tiredness, fever, headache, and stomach ache and there was often no need for medical intervention or disruption to life routine. Furthermore, there were minor adverse events among adolescents (12–15 years old) vaccinated against COVID-19 (441,319 received one dose, 368,228 received two doses, and 102,086 received three doses). Nonetheless, 16 young people experienced myocarditis 5 days after vaccination but recovered 2–6 days after hospitalization without invasive intervention or any specific medication to improve the cardiac function [7].

Parents’ wish to take care of their children’s health, maintain their children’s daily routine regarding their studies (school), and participate in extracurricular activities and extended family gatherings might have influenced the decision to vaccinate their children against COVID-19 [7].

The Arabs are a minority population in Israel, with a lower socioeconomic status than Jews [10]. Over 50% of Israeli Arabs live below the poverty line [11], which may affect the health status of the children.

Rosenstock et al. developed the Health Belief Model (HBM) [12] which relates to several concepts such as perceived benefits of the COVID-19 vaccine and perceived barriers due to the disease (e.g., severity, vulnerability, and threat). On the other hand, the Theory of Reasoned Action (TRA) [13] studies subjective norms and attitudes used to evaluate compliance with vaccination. Both models were the baseline for this study (See Materials and Methods). Among the Israeli Arab society, subjective norms play an important role. In a traditional society where most people live with their relatives in the same village, individuals share their feelings with each other, leading, for example, to most parents wanting to vaccinate their children because other parents have done so. Moreover, attitudes are based on beliefs of the result arising from the behavior of others [14].

Previous studies showed that trust in healthcare authorities was negatively correlated with the willingness of parents to vaccinate their children. Also, pandemic fatigue indirectly affected parents’ decision to vaccinate their children.

The COVID-19 outbreak impacted children physically and mentally. Mandatory isolation of confirmed cases led to a rise in anxiety and depression among those children [15]. Due to the low vaccination rates among the Arab population in Israel [16], which may have dire consequences, it would be interesting to explore Arab parents’ attitudes to the immunization of their younger children and adolescents against COVID-19 and to evaluate the differences between their attitudes towards the two groups.

## 2. Objectives of the Study

To examine factors influencing the willingness of Arab parents to vaccinate their young and/or adolescent children and to assess differences between parents’ attitudes to vaccinate each group.

## 3. Materials and Methods

This cross-sectional study was based on an online survey. The survey was distributed, using the snowball method, to parents of children between the ages of 5 and 18 in the Arab population of Israel through social media networks such as Instagram, Facebook, and WhatsApp. Participants were asked to share the links of the questionnaire to other family members and friends who were parents of children in the relevant age group. The research was approved by the Ethics Committee of Emek Yezreel Academic College (Approval No. 2022-19 YVC EMEK). The data were collected between 22 January and 31 January 2022.

The research questionnaire was based partially on validated questionnaires used In previous studies (see below). The questions were adapted for the current study. The questionnaire included eight parts.

Perceived benefits of the vaccine, perceived barriers to vaccination, and perceived health threats. This questionnaire included 18 questions and was based on the Health Belief Model (HBM) [12].

The perceived benefit of the vaccine was examined by seven items, e.g., “Vaccination against COVID-19 enables children to participate in extracurricular activities as before”, “Vaccination against COVID-19 will help schools operate on a regular basis”.

Perceived barriers to vaccination were examined by six items, e.g., “I am afraid the vaccine will harm my children’s health in the future”.

Perceived health threats consisted of two variables: perceived vulnerability and perceived severity of the illness. The perceived vulnerability was examined by three items, e.g., “The chances my child will contract COVID-19 are high”. Perceived severity of the illness was examined by three items, e.g., “COVID-19 is a dangerous and difficult illness for children”.

Subjective norms and attitudes regarding vaccination compliance. This part consisted of 12 questions constructed according to the Theory of Reasoned Action (TRA) [13,14], e.g., “Most people I know plan to vaccinate their children”. Attitudes regarding the COVID-19 vaccine were assessed by five items, e.g., “I believe that COVID-19 is not dangerous for children, therefore there is no need to vaccinate them”. Respondents were instructed to rank their answers on a 5-point Likert scale, ranging from 1 (do not agree at all) to 5 (agree to a very large extent).

Trust in formal authorities. This part included seven parameters that examined the extent of trust in the authorities (the government, the chief COVID-19 projector, and the Ministry of Health) led the efforts to contain the coronavirus in Israel. Respondents were instructed to respond on a scale of 1 (not at all) to 5 (very much).

Pandemic fatigue. Five statements were constructed based on recommendations of the World Health Organization for formulating policies to cope with pandemic fatigue. Sample items included: “The guidelines issued by various bodies in the government and worldwide contain conflicting messages”. Respondents were instructed to rank their answers on a 5-point Likert scale, ranging from 1 (do not agree at all) to 5 (agree to a very large extent).

Damage caused to children during the pandemic as reported by parents. Five statements were constructed to assess this damage based on a 2020 report by the Myers-JDC-Brookdale Institute [16]. Sample items included: “My child/children spend(s) relatively more time in front of the screen now than before the pandemic” and “My child/children eat(s) more sweets and snacks than before the pandemic”. Respondents were instructed to rate their answers on a scale of 1 (not at all) to 5 (very much).

Parents’ sources of information regarding the vaccine. The questionnaire included five parameters representing different sources of information regarding COVID-19 vaccinations for children, including the Ministry of Education, the Ministry of Health, and social networks such as Facebook and Instagram. Respondents were instructed to respond on a scale of 1 (not at all) to 5 (very much).

Demographic questionnaire. Participants were asked to provide the following personal details about themselves: age, gender, religion, degree of religiosity, marital status, number of children, place of residence (city, village, or other), educational level, occupational status (self-employed, salaried employee, or unemployed), income level, and residential region (north, center, or south).

Vaccination status of parents and children. Parents’ vaccination status was examined by question 10; children’s vaccination status was examined by questions 15 and 18.

Items from the questionnaire written in English were translated into Arabic and back into English by Arab language experts to ensure the reliability of the questionnaire. The introduction to the questionnaire explained the research objectives, promised to maintain respondents’ confidentiality, and stated that participants had the right to respond partially and even to refuse to answer a question or drop out of the study at any time without any consequences.

Data were analyzed using SPSS ver. 28. Demographic and vaccination-related characteristics were described with means and standard deviations as well as with frequencies and percentages. The internal consistency of the scales was assessed with Cronbach’s α. The distribution of the study variables was described with means and standard deviations and Pearson’s correlations between the variables were calculated. Due to the sample size and the multitude of correlation coefficients, the significance level for this specific analysis was set at 0.01. Associations between the demographic variables and the study variables were evaluated by Pearson’s correlations. Demographic variables that were identified with significant associations were controlled for in further analyses. Economic status, assessed by a 5-point scale, had a skewness value of 0.01 (SE = 0.13) and was thus considered a continuous variable. Three groups of parents were defined according to having children and/or adolescents. Differences in vaccination rates between these three parent groups were assessed with the z-ratio for the significance of the difference between independent proportions. Group differences in the study variables between the three groups were evaluated with a multivariate analysis of covariance (MANCOVA) while controlling for parental age and economic status. Estimated marginal means were used to interpret group differences. Two logistic regressions were calculated for child and adolescent vaccination as the dependent variables, with the study variables, while controlling for parental age and economic status. Finally, three vaccination sub-groups were defined among parents who both had younger children and adolescents. Differences in the study variables between these three sub-groups were evaluated with a multivariate analysis of covariance (MANCOVA) while controlling for parental age and economic status. Estimated marginal means were used to interpret group differences.

The Cronbach’s α was calculated for each category of the questionnaire; the value was α = 0.83, α = 0.81, r = 0.56 (*p* < 0.001), α = 0.75, α = 0.87, α = 0.91, α = 0.87, α = 0.89, α = 0.87, α = 0.63, and α = 0.76 for attitudes toward COVID-19 vaccination, subjective norms, perceived susceptibility to the disease, perceived severity of the disease, perceived barriers to vaccination, perceived benefits of vaccination, pandemic fatigue, trust in formal sources, pandemic fatigue, harm caused to children by COVID-19, and for information sources regarding vaccination, respectively.

## 4. Results

Three hundred and sixty-one Arab Israeli parents of children and adolescents participated in the study. They were mostly mothers (about 82%) and about 40 years old on average. Most parents were married (about 97%) and had about three offspring (younger children and/or adolescents) on average. About 52% of the parents had adolescents (age 12–15) and about 84% of them had younger children (age 5–11). Partitioning offspring ages revealed that 130 parents (36.0%) had both adolescents and younger children, 172 (47.7%) had only younger children and 59 (16.3%) had only adolescents.

The parents’ educational levels were rather high. About 26% of them had at least a high school education and the rest had an academic education or were academic students (about 74%). Most lived in northern Israel (about 82%) or in rural areas (about 67%). Most participants were Muslim (about 67%) while the rest were mainly Christian (about 28.5%). Some were religious (about 30%) or partly religious (about 58%). Most were employed (about 69%). The parents had varying degrees of economic status. Almost all their offspring were healthy (Table 1).

The distribution of child vaccination (n = 302) was as follows: one vaccination (n = 21, 7.0%), two vaccinations (n = 35, 11.6%), three vaccinations (n = 1, 0.3%), some children vaccinated and some not (n = 2, 0.7%), and not vaccinated (n = 243, 80.5%). Thus, the one possible categorization was some vaccination (n = 59, 19.5%) vs. not vaccinated (n = 243, 80.5%).

The distribution of adolescent vaccination (n = 189) was as follows: one vaccination (n = 21, 11.1%), two vaccinations (n = 77, 40.7%), three vaccinations (n = 17, 9.0%), some adolescents vaccinated and some not (n = 13, 6.9%), and not vaccinated (n = 61, 32.3%). Thus, to be in line with the children’s categorization, the definition was set as some vaccination (n = 128, 67.7%) vs. not vaccinated (n = 61, 32.3%).

Table 2 presents the vaccination status of the parents and their younger children and adolescents. About 59% of the parents were fully vaccinated as required and about 4% were not vaccinated at all. About two thirds of the adolescents (about 68%) and about one fifth of the younger children (about 20%) had received some vaccination.

Table 3 presents the distribution of the study variables and their correlations. The means for attitudes toward COVID-19 vaccination and subjective norms were a little below the mid-point. Means for perceived susceptibility to the disease, perceived barriers to vaccination, pandemic fatigue, and perceived damage to children by COVID-19 were rather high. The mean for perceived severity of the disease was rather low, while the means for perceived benefits of vaccination, trust in formal sources, and information sources regarding vaccination were a little below the mid-point.

Significant correlations were found between the study variables. Several trends were observed. a. Attitudes toward COVID-19 vaccination, subjective norms, perceived severity of the disease, perceived benefits of vaccination, trust in formal sources, and information sources regarding vaccination were all positively associated with each other. b. Perceived barriers to vaccination and pandemic fatigue were positively associated. c. The first set of variables (a) was negatively associated with the second set (b). d. Perceived susceptibility to the disease and the perception of damage to children by COVID-19 were generally unrelated with the other study variables. e. Child and adolescent vaccination were positively associated with attitudes toward COVID-19 vaccination, subjective norms, perceived severity of the disease, perceived benefits of vaccination, trust in formal sources, and information sources regarding vaccination and were negatively associated with perceived barriers to vaccination and pandemic fatigue.

The associations between the study variables and major demographic variables were assessed to identify demographic variables that should be controlled for in further analyses.

Significant positive correlations were found between parental age and child vaccination, attitudes toward COVID-19 vaccination, subjective norms, perceived severity of the disease, perceived benefits of vaccination, and trust in formal sources (r = 0.16, *p* = 0.002 to r = 0.32, *p* < 0.001). Economic status was positively associated with child vaccination and negatively associated with barriers to vaccination and pandemic fatigue (r = −0.20, *p* < 0.001 to r = −0.26, *p* < 0.001)

Other demographic variables (parent gender, number of children, academic education, residence in rural area, and religion) were not significantly associated with the study variables. Thus, further analyses were performed while controlling for parental age and economic status.

Three parent groups were defined: parents who had only younger children (n = 172, 47.7%), parents who had only adolescents (n = 59, 16.3%), and parents who had both younger children and adolescents (n = 130, 36.0%). The vaccination rate among parents who had only younger children was 16.9% (n = 29) and the vaccination rate among parents who had only adolescents was 79.9% (n = 47), indicating a significant difference (Z = 8.86, *p* < 0.001). Among parents who had both younger children and adolescents, the total vaccination rate was 63.1% (n = 82); among these parents, the vaccination rate for younger children only was 23.1% and that for adolescents only was 62.3%.

Group differences in the study variables (beyond parental age and economic status) were significant only for subjective norms. Parents who had only adolescents reported higher subjective norms (M = 3.11, SD = 0.73) than parents who had only younger children (M = 2.35, SD = 0.76) or those who had both younger children and adolescents (M = 2.56, SD = 0.89) (F (2, 356) = 7.67, *p* < 0.001, η^2^ = 0.041). All other differences were not significant.

Two multiple logistic regression analyses were performed for the study variables, with child and adolescent vaccination as the dependent variables. Parental age and economic status were controlled for, as shown in Table 4. Both models were found to be significant. Child vaccination was mainly associated with attitudes toward vaccination, in addition to perceived barriers to vaccination and perceived benefits. That is, the odds for child vaccination were higher with greater positive parental attitudes and with lower perceived barriers and higher perceived benefits. Adolescent vaccination was associated with perceived barriers to vaccination and perceived benefits. The odds for adolescent vaccination were higher with lower perceived barriers and higher perceived benefits.

As mentioned above, 130 parents had both younger children and adolescents. Partitioning the vaccination rates within this group revealed that 48 parents (36.9%) vaccinated neither their younger children nor adolescents, 52 parents (40.0%) vaccinated only their adolescents, 1 parent (0.8%) vaccinated their younger child, and 29 parents (22.3%) vaccinated both their younger children and adolescents. Thus, three sub-groups were formed: parents who vaccinated neither their younger children nor adolescents, parents who vaccinated only their adolescents, and parents who vaccinated both their children and adolescents. Differences in the study variables between these three groups are presented in Table 5 (controlling for parental age and economic status).

There were significant differences for most variables. The attitudes toward COVID-19 vaccination of parents who vaccinated both their younger children and adolescents were more positive than the attitudes of parents who chose to vaccinate only their adolescents. The attitudes of both parent groups were more positive than the attitudes of parents who vaccinated neither their younger children nor adolescents. The influence of subjective norms was higher for parents who vaccinated both their younger children and adolescents than for parents who vaccinated neither.

The severity of the disease was perceived as higher by parents who vaccinated both their younger children and adolescents than by the other parents. Barriers to vaccination were perceived as highest by parents who vaccinated neither their younger children nor adolescents, as lower by parents who vaccinated only their adolescents, and as lowest by parents who vaccinated both their younger children and adolescents. Similarly, the benefits of vaccination were perceived as highest by parents who vaccinated both their children and adolescents, as lower by parents who vaccinated only their adolescents, and as lowest by parents who vaccinated neither their younger children nor adolescents.

Pandemic fatigue was higher for parents who vaccinated neither their younger children nor adolescents than for all other parents. Trust in formal sources was higher among parents who vaccinated both their younger children and adolescents than among the other parents. Finally, the use of information sources regarding vaccination was higher among parents who vaccinated their younger children and/or adolescents than among parents who vaccinated neither.

All in all, as expected, parents who chose to vaccinate their younger children and/or adolescents reported more favorable perceptions of the vaccination than parents who chose not to vaccinate their younger children and/or adolescents. In some cases, parents who vaccinated both their younger children and adolescents showed more favorable perceptions than parents who chose to vaccinate only their adolescents.

## 5. Discussion

In our study, 80% of the young children and 32% of adolescents were not vaccinated compared to 38% and 33% in the general population, respectively (the Israeli MOH statistics on 2022) [17]. Hence, the vaccination rate of young children is much lower among the Arab population than that among the general population, while that of adolescents was similar.

Most of our participants were mothers and had over 12 years of education. A previous study showed that education was positively correlated with higher rates of vaccination among Arab adults [18]. In the current study, education was not significantly associated with the rate of sibling vaccination. While some studies have shown a negative correlation between parents’ educational level and intention to vaccinate young [19], others have reported a positive correlation between the same variables [20]. We may conclude that there is no consistency in the relationship between parents’ level of education and intention to vaccinate their children, or there may be some other factors that might have influenced the role of educational level in parents’ decision to vaccinate their children.

Furthermore, while 59% of parents were fully vaccinated (four doses) and only 4% were not vaccinated, most of the young children in the study (80.5%) and 32.3% of adolescents were not vaccinated. So, there is a wide difference between the vaccination rate of parents and that of their children. Even among siblings in the same families, there were differences between the vaccination rates of adolescents and younger children. In this study, 130 parents had both younger children and adolescents. Partitioning the vaccination rates within this group revealed that 48 parents (36.9%) chose not to vaccinate both their younger children and adolescents while 52 parents (40.0%) chose to vaccinate only their adolescents. Therefore, it seems that parents choose to vaccinate themselves against COVID-19 for several reasons while opting not to vaccinate their adolescents and/or younger children. Below are some possible explanations for such a decision:Age eligibility: The COVID-19 vaccines initially received emergency use authorization for adults, with vaccination campaigns prioritizing older individuals and those at higher risks. As a result, parents were eligible for vaccination before their children. Furthermore, understanding the mortality rate of the disease for certain populations revealed that the virus is more harmful to older people and to those with certain illnesses like cardiac, malignant, and lung diseases and diabetes. If young people were more vulnerable to the virus, clinical trials of the vaccine would have started with a younger cutoff [21].Vaccine availability: Vaccine distribution and availability varied across different regions and countries. It is possible that parents were able to access the vaccine sooner than their children due to logistical factors or variations in vaccine rollout plans. In Israel, younger people, at the early stages of the vaccine rollout, could get vaccinated only if they were at a higher risk of infection, e.g., if they were volunteers in health systems or severely ill. Healthy adolescents and young children who were not at a high risk of infection were the last to get vaccinated because of the shortage of vaccines during the first stages of the pandemic. The vaccines were not distributed to all countries evenly and wealthier countries purchased the vaccines before others [22]. In Israel, there was a shortage of vaccines at the first stages of the pandemic (physical boundaries). Furthermore, like we have shown in a previous study, in the early stages of vaccination, access to the vaccine among the Arab adults was lower than that among the general community in rural areas in Israel [18]. However, the coverage of vaccination of Arab children was like that among the general population and there were no physical boundaries to get the vaccine;Risk assessment: Parents might have assessed the risks and benefits of vaccination differently for themselves and their children. COVID-19 tends to pose a higher risk to older individuals or those with underlying health conditions. Parents probably perceived their own risk of contracting severe diseases to be higher than that of their siblings and, therefore, might have prioritized getting vaccinated to protect themselves. During the pandemic, children had lower risks of developing severe illnesses and some who were infected with the virus were quite asymptomatic. Furthermore, parents probably did not consider that these asymptomatic children could infect older ones in the family. This somehow, in our study, may explain why older parents tended to vaccinate their children more than younger ones, as was shown in some other studies [23]. Since Israeli Arabs live in closed communities, we had expected that they would show more concern for their elderly. However, other cultural factors specific to the community might have played a role, as Al-Ghuraibi et al. found among Arabs in Saudi Arabia [24];Vaccine hesitancy or concerns: Following the introduction of the vaccine to adolescents and children, there was some hesitancy among the general population as to whether to vaccinate children and adolescents as the disease symptoms were less significant among the young population. On the other hand, infected children could be a risk factor in infecting the older population with the new variants. Furthermore, there was an argument as to whether this young population was in need of vaccination as there was a decline in the number of severe cases among the older population. In addition, parents were perhaps awaiting longer-term safety data, seeking additional medical advice, or considering factors like the child’s age, health condition, or potential side effects of the vaccine. Parents might have felt more confident about getting vaccinated themselves but they were more cautious about vaccinating their children;Trust in the vaccines: Individual attitudes toward vaccination can vary. Parents might have shown more trust in the safety and efficacy of the COVID-19 vaccines for themselves, based on research, personal experiences, and/or conversations with healthcare professionals. However, they apparently had reservations or uncertainties about vaccinating their children, requiring further information or reassurance before proceeding. Yet, among the Bedouin Arabs in Israel, the authorities had difficulties in vaccinating children against common infant viruses (like polio) due to physical barriers and the government’s neglect of this Arab community [25];Pandemic fatigue: Studies have revealed that pandemic fatigue was positively correlated with low compliance with safety instructions and reluctance to get vaccinated. The longer the pandemic lasted, the more anxious and exhausted people became, leading to indifference to the risks of the virus. This may partially explain the lower percentage of young children who were vaccinated compared to adolescents as the vaccination of young children happened toward the end of the pandemic [26];Psychological subjective barriers and social norms: Although not all Israeli Arabs live in rural areas, they still live in communities in towns where social norms play a role in their attitude toward vaccinating their children. Social norms were shown to affect the willingness of adults to be vaccinated at the beginning of the pandemic [27]. In closed communities, social norms may play significant roles in choosing to vaccinate against COVID-19. The strength of the relationship tends to decline as the queried social group grows larger and indeed so are the effects of social norms on the vaccination of children [28];Social media networks (SMN): In this pandemic, social media played a significant influence on people’s perception of the severity of disease, the reliability and dangers of vaccination, and there was a great competition between the official media and the SMN. We are facing a new era were SMN and faked news or videos may influence peoples’ decisions very badly [29]. In a previous study among Israeli Arabs, at the beginning of the vaccinations of the adults, we witnessed a great deal of misinformation published in the SMN, some of which were arguably unreasonable. Many Arabs positively responded to a question in our study that addresses the fear of Arabs from government’s intention to use the vaccination as a tool through which to inject a chip and follow Arab minority movements in the country [18]. However, it took a while until vaccinations reached young children and this misinformation subsided. Yet, the SMN continued to be very active and an argument was raised against vaccinating young children as it was not proved safe enough. This might have added to the whole population’s hesitancy as well as to that of Arabs citizens who live mainly in smaller communities and share misinformation very easily. However, since Arabs in Israel are a minority living in different demographic characteristic from Arabs in neighboring Arab countries, the authors did not find a place to compare the results with other countries in the region.

Significant positive correlations were found between parental age and child vaccination, attitudes toward COVID-19 vaccination, subjective norms, perceived severity of the disease, perceived benefits of vaccination, and trust in formal sources. These findings correlate well with previous studies among the adult Arab population [18] as well as among other nations worldwide [30]. Since symptoms of the disease were severe among the elderly, this might have influenced the perceived severity of the disease among older parents and the higher willingness to vaccinate their children.

Parents who had only adolescents reported significantly higher subjective norms than parents who had only younger children or those who had both younger children and adolescents. Perhaps this was influenced by the higher hesitancy concerning the vaccine’s safety and need among children as symptoms were also milder. Furthermore, the odds for child vaccination were higher with greater positive parental attitudes and with lower perceived barriers and higher perceived benefits. This point is worth being considered by authorities so they may invest more in a campaign of explanation and delivering clear and precise information to parents, prevent or deal with misinformation, and increase trust between both parties.

## 6. Conclusions

It is important to note that family, community, cultural, and psychosocial backgrounds may affect peoples’ decisions to vaccinate themselves and their children. The reasons behind a parent’s decision to vaccinate themselves but not their adolescent and/or younger children may vary significantly. Many studies have analyzed the reasons why people resisted COVID-19 vaccination and how cultural backgrounds affect different sectors of a country. In the event of another pandemic, public health authorities and healthcare providers must promptly address people’s concerns, provide accurate information, and address low vaccine uptake among minorities, if that is the case, in order to protect the health of all citizens. The authorities should also move to prevent people from buying into false news on social media.

In the context of Arabs as a minority, our government, as well as some other governments, should invest more effort in addressing minorities’ needs because minorities are not presented enough, and hence, there may be some psycho-social and physical boundaries in their way to being vaccinated. Although most Arabs are speakers of the Arabic and Hebrew languages, it took a while until authorities realized that an additional campaign should be held in Arabic. In pandemics, the authors believe that “the strength of a chain is as strong as the weakest link within it”. A continuous effort should be made regardless of pandemics, where a lack of continuous effort will come back as a boomerang toward the whole population. Pandemics are the proper case for governments to internalize this point.

## 7. Limitations of the Study

In this study, the distribution of the questionnaire could not be well controlled as it was distributed via social media. Most of the respondents were mothers. Thus, we cannot estimate the number of fathers who decided not to respond to the questionnaire, therefore, we cannot estimate their willingness to vaccinate their children. However, we assume that the bottom decision and final results reflect both parents’ decision to vaccinate both children and adolescents in each family.

Other factors may influence parents to vaccinate their children. Regarding COVID-19 vaccination, needle phobia might have contributed to hesitancy or resistance in children. It is estimated that needle phobia affects 10% in a study conducted in the UK [31]. The fear of pain or discomfort associated with needles may be higher among children or among parents who consider this fear more important than the need for COVID-19 vaccination for children. Parents might have considered this fear and indulged their children’s wish not to get vaccinated because of the fear. This phobia may be found to a lesser degree among teenagers, although it may still play a role in their parents’ decisions to vaccinate them. In our study, we did not anticipate the results we obtained; therefore, we did not examine this point. This factor of needle phobia is worth examining in future studies. Additional factors that were not measured here, such as parent–child relationships, the marital relationships, or the family’s social networks, might have affected the participants’ willingness to be vaccinated. These factors, and the willingness to be vaccinated itself, might have changed over time as well. Thus, future studies are advised to look at such additional factors and the willingness to be vaccinated, over time. Finally, our last analysis compared three sub-groups of parents having both children and adolescents, with a total n of 129. This is a small sub-sample, the results of which should be regarded with caution. A power analysis of these results showed that for the significant result with the lowest effect size (η^2^ = 0.068) power was moderate, i.e., = 0.78 and all other power values for significant results were above 0.85. Thus, 8 of the 10 comparisons in the table have sufficient levels of power. Still, as groups were not randomly sampled, results should be regarded with caution and future studies are advised to look at larger sub-groups.

## Figures and Tables

**Table 1 vaccines-11-01540-t001:** Participants’ sociodemographic characteristics (n = 361).

Characteristics	
Mean age, years (SD), range	39.45 (6.33), 25–59
Gender, female, n (%)	297 (82.3%)
Family status, married, n (%)	351 (97.2%)
Mean number of children (SD), range	3.14 (1.18), 1–9
Adolescent offspring (age 12–15), yes, n (%)	189 (52.4%)
Child offspring (age 5–11), yes, n (%)	302 (83.7%)
Education, n (%)	
High school or less	55 (15.2%)
Above high school	40 (11.1%)
B.A. (or a B.A. student)	127 (35.2%)
M.A. student	42 (11.6%)
M.A., Ph.D	97 (26.9%)
Area of living, northern Israel, n (%)	295 (81.7%)
Type of living, rural, n (%)	242 (67.0%)
Religion, n (%)	
Muslim	242 (67.0%)
Christian	103 (28.5%)
Druze	14 (3.9%)
Other	2 (0.6%)
Religiosity, n (%)	
Secular	42 (11.6%)
Partly religious	210 (58.2)
Religious	109 (30.2)
Employment, yes, n (%)	250 (69.3%)
Economic status, n (%)	
Below average	126 (34.9%)
Average	140 (38.8%)
Above average	95 (26.3%)
Offspring’s health, all healthy, n (%)	350 (97.0)

**Table 2 vaccines-11-01540-t002:** Vaccination status of the parents and their offspring (n = 361).

Characteristics	
Parent vaccination, n (%)	
Three or four vaccinations	213 (59.0%)
Two vaccinations, was sick, or was sick and one vaccination	134 (37.1%)
No vaccination	14 (3.9%)
Adolescent vaccination, (n = 189), n (%)	
Yes, one to three vaccinations	128 (67.7%)
No vaccination	61 (32.3%)
Child vaccination, (n = 302), n (%)	
Yes, one to three vaccinations	59 (19.5%)
No vaccination	243 (80.5%)

**Table 3 vaccines-11-01540-t003:** Means, standard deviations, and correlations for the study variables (n = 361).

	M (SD)	1.	2.	3.	4.	5.	6.	7.	8.	9.	10.	11.	12.
1. Child vaccination	0.20 (0.40)	1											
2. Adolescent vaccination	0.68 (0.47)	0.39 **	1										
3. Attitudes	2.34 (1.12)	0.62 **	0.42 **	1									
4. Subjective norms	2.55 (0.84)	0.32 **	0.31 **	0.49 **	1								
5. Susceptibility	3.90 (1.00)	−0.01	−0.01	−0.15 *	−0.02	1							
6. Severity	2.29 (0.90)	0.37 **	0.34 **	0.56 **	0.51 **	0.01	1						
7. Barriers	3.62 (1.03)	−0.58 **	−0.47 **	−0.70 **	−0.48 **	0.04	−0.45 **	1					
8. Benefits	2.82 (1.11)	0.49 **	0.45 **	0.55 **	0.56 **	−0.01	0.49 **	−0.54 **	1				
9. Pandemic fatigue	3.94 (0.84)	−0.24 **	−0.23 *	−0.43 **	−0.26 **	0.13	−0.34 **	0.49 **	−0.30 **	1			
10. Trust—formal	2.64 (0.89)	0.28 **	0.20 *	0.30 **	0.34 **	−0.01	0.35 **	−0.31 **	0.46 **	−0.43 **	1		
11. COVID harmed child	3.64 (0.80)	0.01	−0.01	−0.08	−0.06	0.18 **	−0.02	0.05	0.14 *	0.22 **	−0.02	1	
12. Information sources	2.85 (0.91)	0.18 *	0.18 *	0.18 **	0.37 **	0.14 *	0.28 **	−0.23 **	0.39 **	−0.27 **	0.50 **	0.08	1

* *p* < 0.01, ** *p* < 0.001. Adolescent and child vaccination: range 0–1. All other variables: range 1–5. Due to the sample size, the significance level was set at 0.01.

**Table 4 vaccines-11-01540-t004:** Logistic regressions for child and adolescent vaccination.

	Child Vaccination (n = 302)			Adolescent Vaccination (n = 189)		
	B (SE)	OR (95% CI)	*p*	B (SE)	OR (95% CI)	*p*
Parent age	0.01 (0.04)	1.00 (0.93, 1.08)	0.923	0.03 (0.03)	1.03 (0.97, 1.09)	0.375
Economic status	0.01 (0.21)	1.00 (0.67, 1.5)	0.999	0.07 (0.14)	1.08 (0.82, 1.41)	0.601
Attitudes	1.24 (0.30)	3.47 (1.93, 6.24)	<0.001	0.36 (0.26)	1.43 (0.86, 2.36)	0.164
Subjective norms	−0.40 (0.36)	0.67 (0.33, 1.35)	0.260	−0.17 (0.35)	0.85 (0.42, 1.69)	0.639
Susceptibility	0.21 (0.24)	1.23 (0.77, 1.98)	0.384	0.20 (0.20)	1.22 (0.83, 1.8)	0.319
Severity	0.11 (0.31)	1.11 (0.61, 2.04)	0.727	−0.10 (0.31)	0.91 (0.50, 1.66)	0.748
Barriers	−0.73 (0.31)	0.48 (0.26, 0.89)	0.019	−1.26 (0.36)	0.28 (0.14, 0.57)	0.000
Benefits	0.63 (0.30)	1.88 (1.04, 3.38)	0.036	0.90 (0.27)	2.45 (1.45, 4.14)	0.001
Pandemic fatigue	0.35 (0.32)	1.42 (0.75, 2.66)	0.279	0.28 (0.32)	1.32 (0.70, 2.48)	0.386
Trust—formal	0.45 (0.36)	1.56 (0.77, 3.19)	0.220	−0.36 (0.27)	0.70 (0.41, 1.19)	0.186
COVID harmed child	0.13 (0.29)	1.14 (0.64, 2.02)	0.656	−0.24 (0.28)	0.79 (0.45, 1.36)	0.390
Information sources	0.03 (0.36)	1.03 (0.51, 2.08)	0.928	0.15 (0.25)	1.16 (0.71, 1.89)	0.545
Nagelkerke’s R^2^	0.605			0.445		
χ^2^ (12)	141.08, *p* < 0.001			71.60, *p* < 0.001		

**Table 5 vaccines-11-01540-t005:** Means, standard deviations, and F values for the study variables by the vaccination sub-groups of parents having both children and adolescents (n = 129).

	No Vaccination of Both Child and Adolescent (n = 48) M (SD)	Adolescent Vaccination (n = 52) M (SD)	Child and Adolescent Vaccination (n = 29) M (SD)	F(2, 124) (*p*) (η^2^)
Attitudes	1.76 _c_ (0.89)	2.22 _b_ (0.90)	3.59 _a_ (0.91)	35.58 (*p* < 0.001) (η^2^ = 0.365)
Subjective norms	2.23 _b_ (0.88)	2.57 _ab_ (0.83)	3.09 _a_ (0.78)	8.18 (*p* < 0.001) (η^2^ = 0.117)
Susceptibility	3.75 (1.21)	3.83 (1.02)	3.66 (1.08)	0.14 (*p* = 0.869) (η^2^ = 0.002)
Severity	1.90 _b_ (0.80)	2.27 _b_ (0.86)	2.90 _a_ (0.86)	11.37 (*p* < 0.001) (η^2^ = 0.156)
Barriers	4.30 _a_ (0.64)	3.67 _b_ (0.96)	2.57 _c_ (0.92)	34.74 (*p* < 0.001) (η^2^ = 0.361)
Benefits	2.22 _c_ (0.92)	2.88 _b_ (1.05)	3.72 _a_ (0.71)	21.96 (*p* < 0.001) (η^2^ = 0.263)
Pandemic fatigue	4.29 _a_ (0.66)	3.88 _b_ (0.85)	3.71 _b_ (0.81)	5.57 (*p* = 0.005) (η^2^ = 0.083)
Trust—formal	2.31 _b_ (0.89)	2.54 _b_ (0.75)	3.07 _a_ (0.93)	6.30 (*p* = 0.002) (η^2^ = 0.093)
COVID harmed child	3.70 (0.89)	3.69 (0.68)	3.59 (1.01)	0.18 (*p* = 0.832) (η^2^ = 0.003)
Information sources	2.58 _b_ (0.94)	3.01 _a_ (0.75)	3.14 _a_ (0.94)	4.38 (*p* = 0.015) (η^2^ = 0.068)

Note. Different letters mark significant differences.

## Data Availability

All data generated or analyzed during this study are included in this published article.

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
