# Peer review of "Factors Related to the Compliance of Arab Parents in Israel to the Vaccination of Children and Adolescents against COVID-19"

_vaccines, 2023, doi:10.3390/vaccines11101540_

Round 1
Reviewer 1 Report
This is an excellent paper highlighting factors related to the compliance of Arab parents in Israel to the vaccination of children and adolescents against Covid19.
80.5% of Arab children had very no vaccinations. It would be great if this study could be expanded, so the results are generalisable to other marginalised rural populations. What was the uptake within the general population?
Author Response
Reviewer #1:
Thank you very much for your comment. According to your suggestion we added information about the rate of vaccination of children and adolescents in the general population. See please the discussion. line 327
There was another group that behaved like the Arab minority even with lower rate of vaccination, and that is the ultra-orthodox Jews, however we preferred not to deal with this group in our study.
Reviewer 2 Report
A very well conducted study. The results are more interesting regionally than globally. It would be advisable to repeat the study for both adults and children, now that the world knows much more about the course of the disease and vaccinations.
Author Response
Thank you very much for your comment and suggestion. Unfortunately, we did not find researches dealing with adolescents and young children rate of vaccination in other Arab countries in the region. However, we added a paragraph about this point in other countries in the near east, and we discussed the special situation of Arabs in Israel as a minority compared to Arab citizens in Arab countries in the area. Yet, it would be very interesting investigate this data. See line 432
Reviewer 3 Report
You have investigated mostly mothers (82,3%). What was the role of fathers in decision to vaccinate children/adolescents - the word of a father is traditionally decisive in Arabic population?
What is children/adolescents general vaccination coverage of the Arabic minority?
You should characterize much more the vaccine hesitancy of Arabic minority parents.
You should prove the injection fear in your study.
What was the role of misinformation about COVID-19 vaccines/vaccination?
What are the implications of your study results for public health?
Author Response
Thank you very for your valuable notes!
Comment #1: definitely you are right. Only 18% of respondents to the questionnaire were fathers. However, we assume that the results and the percent of vaccinated children and adults reflect the decision of both parents, and that is the main point. After you comment we added a paragraph that addresses this point in the limitation of the study. See paragraph line 470
Comment #2: the coverage of the vaccination of Arabs children was the same as that among all Israeli population and there were no physical boundaries as it was among the adults at the beginning of the vaccination process. See lines 370.
Comment #3: following your comment, we added a paragraph that explains Arab parents' hesitancy to vaccinate their children in the context of the whole population hesitancy. (Lines 386 – 396). We hope this meets your expectations.
Comment #4: you are right, however, we did not include this question in the questionnaire as we did not expect to get these results. Therefore, on discussing the results we searched the literature and we found many who wrote about this point, and we added a paragraph with a reference in the limitations of the study line 476 - 485.
Comment #5: thank you for your comment. We added another paragraph under "social media networks" detailing our experience in a previous study that examined this point (lines 419 – 434).
Comments #6: it seems that the implication of our study was not written clear enough. Therefore, we added another paragraph under conclusions. Hope now it is clearer. We believe minorities are not considered or nourished enough by governments all along the time, and this point may cause harm to the entire country during pandemics. See please paragraph 461 – 469.
Reviewer 4 Report
The topic selection of this paper has certain research value and has obtained some valuable results, but there are several problems in the following aspects:
1. Are there any relevant research topics from other countries or regions at present? If so, what are the advantages of this study? Suggest adding explanations in the preface section.
2. Is the population surveyed on the internet sufficiently representative? The randomness of snowball sampling is relatively poor, why not choose random sampling?
3. How is it considered that the sample size included in the final analysis is relatively small, which may lead to insufficient statistical significance of important factors related to vaccination willingness?
4. There are many factors that affect the willingness of the population to receive vaccines, and over time, the willingness may also change, which may affect the results of this study.
5. The analysis and comparison of the results during the discussion were not in-depth enough to reflect the value of the results of this study
Moderate editing of English language required
Author Response
First, we would like to thank you for your comments, as it gave us the opportunity to make the manuscript clearer.
Comment #1: there are some researches from neighboring Arab countries regarding hesitancy of adults to get vaccinated. We discussed this point in a previous study (Reference no 18). However, there were no researches dealing with children. Furthermore, the authors did not find place to make the comparison again, as Arabs in Israel are a minority with political issues and mistrust with authorities that does not exist among Arab neighbor countries. We added a paragraph according to your comment line 432- 434.
Comment #2: thank you for your note. The advantage of the snowball method is reaching more participants, however, we can`t control the participation of equal numbers of fathers and mothers. We assume that the decision to vaccinate children was not done by one parent unless there is a single parent. Yet, following your comment we added a paragraph dealing with this point in the limitations of the study (Line 470)
Comment #3: Sample size in the final analysis was small. We have added a limitation in that regard in the final section of the discussion. Further, a power analysis of the results shown in table 5 was calculated and revealed that most comparisons had a sufficient level of power. Still, because sampling was not random we advised that the results should be regarded with caution, and recommended that future studies look at larger sub-groups. (Line 485 – 497)
Comment #4: Indeed, additional factors, which were not measured in our study, might have affected the participants’ willingness to be vaccinated. These factors, and the willingness to be vaccinated itself, might have changed over time as well. We have added a limitation in that regard in the final section of the discussion. (Line 485 – 497)
Comment #5: following your comment, we detailed our discussion to include most results, to compare to what is written in others` studies, and try to explain the findings or suggest further investigation and recommendations to authorities. (Lines 435 – 450) and added table 5 to better clarify the results. Thank you!